# CURE: Concept Unlearning via Orthogonal Representation Editing in Diffusion Models

**Shristi Das Biswas**[*], **Arani Roy**[*], **Kaushik Roy**
Purdue University
{sdasbisw, roy173, kaushik}@purdue.edu

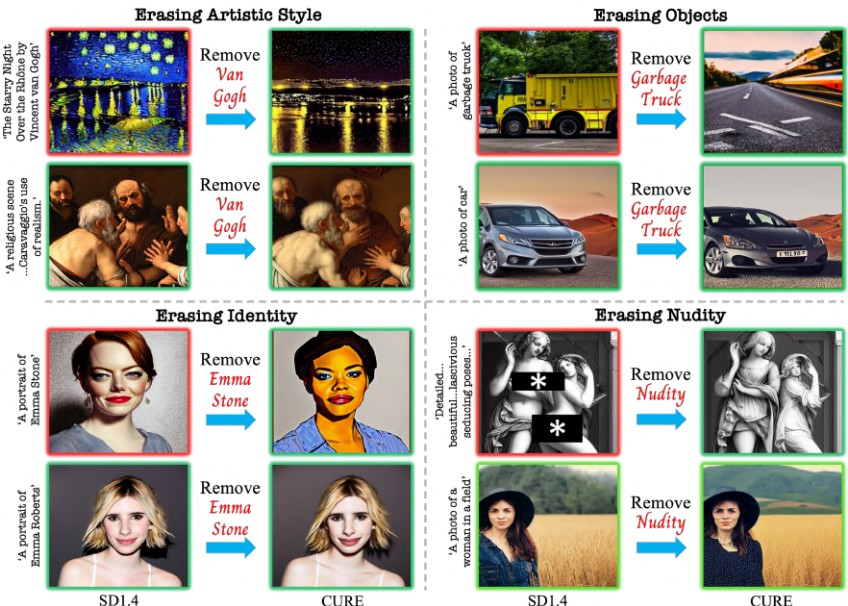

Figure 1: Our method, CURE, enables robust and efficient erasure of any target concept in text-to-image models through orthogonal closed-form editing of cross-attention weights, ensuring that the unintended concepts remain intact, even if they share common terms with the target concept (seen in the bottom-left sample). This can safeguard celebrity portrait rights, respect copyrights on artworks, and prevent explicit or unwanted content creation in a training-free manner with high efficacy.

## Abstract

As Text-to-Image models continue to evolve, so does the risk of generating unsafe, copyrighted, or privacy-violating content. Existing safety interventions - ranging from training data curation and model fine-tuning to inference-time filtering and guidance - often suffer from incomplete concept removal, susceptibility to jail-breaking, computational inefficiency, or collateral damage to unrelated capabilities. In this paper, we introduce CURE, a training-free concept unlearning framework that operates directly in the weight space of pre-trained diffusion models, enabling fast, interpretable, and highly specific suppression of undesired concepts. At the core of our method is the Spectral Eraser, a closed-form, orthogonal projection module that identifies discriminative subspaces using Singular Value Decomposition over token embeddings associated with the concepts to forget and retain. Intuitively, the Spectral Eraser identifies and isolates features unique to the undesired concept while preserving safe attributes. This operator is then applied in a single step update to yield an edited model in which the target concept is effectively

39th Conference on Neural Information Processing Systems (NeurIPS 2025).

unlearned - without retraining, supervision, or iterative optimization. To balance the trade-off between filtering toxicity and preserving unrelated concepts, we further introduce an Expansion Mechanism for spectral regularization which selectively modulates singular vectors based on their relative significance to control the strength of forgetting. All the processes above are in closed-form, guaranteeing extremely efficient erasure in only 2 seconds. Benchmarking against prior approaches, CURE achieves a more efficient and thorough removal for targeted artistic styles, objects, identities, or explicit content, with minor damage to original generation ability and demonstrates enhanced robustness against red-teaming. Project Page at `https://sites.google.com/view/cure-unlearning/home`.

# 1   Introduction

Recent Text-to-Image (T2I) diffusion models have garnered significant attention for their ability to synthesize high-quality, diverse images across a wide range of prompts (1; 2; 3; 4; 5; 6). These capabilities arise from training on large-scale, uncurated internet datasets (7), which inadvertently expose models to undesirable concepts (8), such as copyrighted artistic styles (9; 10; 11; 12), deepfakes (13; 14) or inappropriate content (15; 16). Such risks underscore the need for principled concept unlearning solutions to eliminate harmful, copyright-enforced, or offensive knowledge, enabling a safer and more responsible deployment of generative models.

A natural first step in improving safety in generative models is to curate training data to exclude undesirable content (17). However, retraining large models and re-annotating datasets to meet evolving safety standards is prohibitively expensive (18), and data filtering alone often leads to unintended consequences: removing one type of undesired content can expose other undesired content (19), introduce new biases (20), or result in incomplete removal (21), highlighting the limitations of data curation alone. To tackle these challenges, recent research has incorporated safety mechanisms into diffusion models. Safeguarding methods apply safety checkers to censor outputs (22), or steer generation away from unsafe concepts using classifier-free guidance and prompt filtering (16; 23). However, these approaches introduce recurring computational overheads as guardrails must be applied for every new prompt at runtime, leading to quality degradation in cases of prompts subjected to hard filtering due to distribution shift. Furthermore, they are easily circumvented in open source settings where model code and parameters are publicly accessible (24; 25). In response to the drawbacks mentioned above, an alternative is to unlearn undesirable concepts from T2I models by fine-tuning their parameters (26; 27; 28; 29; 30; 31). Given an undesirable concept, these methods aim to prevent the generation of undesirable content by updating the model's internal representations. Compared to full retraining or inference-time interventions, unlearning concepts by removing their knowledge from the model weights offers a more efficient and tamper-resistant solution, especially in open-source settings. However, most existing approaches require numerous fine-tuning iterations over large parameter subsets (27; 28; 29), leading to substantial computational cost and degradation in the model's general generation quality. A compelling solution to address the above issues is offered by model editing frameworks (32; 33; 34; 35) which modify model weights in closed-form to enhance safety without the previous overheads of gradient-based fine-tuning. While more efficient, these methods often fail to completely suppress targeted inappropriate concepts, remaining vulnerable to adversarial prompts uncovered through red-teaming (36; 37; 38), which can trigger the regeneration of supposedly 'forgotten' content in the unlearned model. Thus, there is an urgent need for an efficient and reliable mechanism to ensure safe visual generation across a wide range of contexts.

Motivated by the geometric relationship between embedding and weight spaces in diffusion models, this paper presents CURE, a mathematically principled and efficient framework for reliably erasing undesired concepts in a single-step update. Building upon prior efficient concept unlearning methods that apply closed-form solutions to modify diffusion model layers, CURE advances these approaches by explicitly exploiting orthogonal geometric structures derived through Singular Value Decomposition (SVD) (39; 40). Specifically, we propose the Spectral Eraser, a closed-form spectral operator that constructs discriminative subspaces by decomposing token embeddings into orthogonal subspaces associated with concepts designated for removal, 'forget concepts,' and those intended to be preserved, 'retain concepts'. By translating the derived embedding-space projections directly into weight-space modifications, the Spectral Eraser precisely removes directions uniquely tied to harmful or undesired concepts while preserving the semantic integrity of overlapping and unrelated

representations. Furthermore, to flexibly control the extent of concept removal, we integrate a singular-value expansion strategy, inspired by Tikhonov regularization from classical inverse problems (41; 42; 43). This mechanism selectively scales singular vectors based on their normalized spectral energies, emphasizing or de-emphasizing the discriminative-ness of the derived forget and retain subspaces. Consequently, such spectral regularization achieves a mathematically interpretable and precise trade-off between effective concept removal and preservation of unrelated capabilities. All updates within CURE are performed in closed-form, ensuring exceptional computational efficiency, with robust concept removal in approximately 2 seconds. Our contributions are summarized as:

- We present CURE, a strong, scalable and training-free concept unlearning method leveraging orthogonal projections and spectral geometry, to yield a closed-form weight update operator, dubbed Spectral Eraser, for reliable and responsible visual content creation in T2I models.
- CURE introduces a selective singular-value Expansion Mechanism, grounded in classical regularization theory, to balance robust concept suppression and semantic preservation.
- Extensive experiments demonstrate that CURE effectively and robustly removes unsafe content, artist-specific styles, object and identities. It significantly outperforms existing training-based and training-free methods in terms of generation quality, efficiency, specificity, and resistance to adversarial red-teaming tools for both single- and multi-concept removals.

## 2 Related Works

**Concept Unlearning**  Existing approaches for removing undesired concepts in T2I models primarily fall into three distinct categories. The first class focuses on **post-hoc, inference-time control**, leveraging safety checkers (22) or classifier-free guidance during generation (16; 44). Similarly, (23) operates without modifying weights, filtering the embeddings away from identified unsafe subspaces and attenuating harmful latent features during denoising. However, these soft intervention strategies can be easily circumvented by malicious users in open-source settings where the model architectures and parameters are publicly available (24) and require independent safeguarding operations for every new prompt, hindering inference-time efficiency. The second category involves **training-based interventions** (45; 46; 47; 48), where the model is retrained on filtered datasets, finetuned using negative guidance (30; 27; 49; 26), or makes attempts to minimize the KL divergence between unwanted and alternative safe concepts (29) to suppress unsafe generations. Adversarial training frameworks such as (50) neutralize harmful text embeddings while works like (51; 52) remove unwanted representations through preference optimization. Latent space manipulation, explored by (53; 54) enhance safety using self-supervised learning. On the other hand, partial parameter finetuning approaches adjust specific layers to forget or suppress undesired concepts (31; 28). Although effective in removing specific knowledge from the model, these methods are computationally expensive, require extensive data curation, cause performance degradation, and are easily bypassed by red-teaming tools for T2I diffusion models (55). Combining the benefits of the earlier categories is the third direction proposing **training-free strategies** that adjust model behavior without retraining to erase specific concepts from model weights. Techniques like (35; 34) perform projection-based model editing in the attention layers whereas (32) applies minimal parameter updates to diffusion models to remove harmful content while preserving generative capacity. However, these methods often lack robustness against adversarial prompts and fail to show persistent unlearning. Taking inspiration from class unlearning and continual learning approaches for non-generative models that have leveraged the benefits of orthogonal representations to remove and add tasks to a model's learnt representation, respectively (56; 57), we formulate our approach in a similar stride, exploring orthogonal concept erasure in T2I models to guarantee efficient unlearning.

**Red-Teaming Attacks for T2I Diffusion Models**  As safety mechanisms become more prevalent, recent works have explored adversarial attacks (58; 59) and jail-breaking (60) to evaluate the robustness of unlearned T2I models. White-box attacks like (38; 61; 36) exploit the classification capacity or prompt-conditioned behavior of diffusion models to revive erased concepts. In contrast, black-box methods like (37) use evolutionary algorithms to generate adversarial prompts or exploit text embeddings and multimodal inputs to bypass safeguards (62). These tools reveal critical vulnerabilities in concept removal approaches when deployed in unrestricted environments and while several unlearning frameworks partially mitigate these attacks, very few are robust across all threat types.

Unlike previous works that fail to satisfy the tri-fold requirement of efficient unlearning, persistent erasure against adversarial prompts and preservation of generation quality, this paper bridges the gap

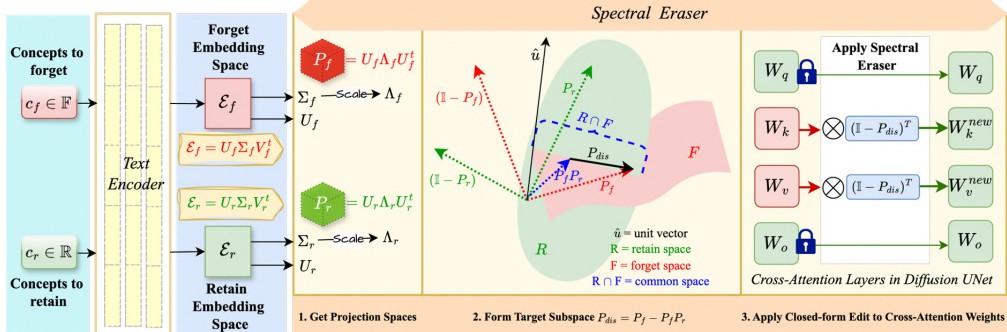

Figure 2: Overview of CURE. Given forget ($\mathcal{F}$) and retain ($\mathcal{R}$) sets, CURE constructs energy-scaled projectors ($\mathcal{P}_f$, $\mathcal{P}_r$) over their respective subspaces (Part 1), and derives a composite projection $\mathcal{P}_{\text{dis}}$ to suppress erasable components while preserving shared content (Part 2). The action of $\mathcal{P}_f$, $\mathcal{P}_r$, and $\mathcal{P}_f \mathcal{P}_r$ on unit vectors yields directions aligned with the forget (red), retain (green), and shared (blue) subspaces. This operator is applied to cross-attention (Part 3), enabling lightweight unlearning.

by introducing CURE, a single-step, training-free weight update framework that applies a closed-form Spectral Eraser to isolate and remove undesired knowledge from diffusion models using tunable unlearning strength to guarantee robust, interpretable and efficient removal of any target concept(s).

## 3 Method

We present CURE, a training-free approach to concept removal in pretrained diffusion models that operates by constructing tunable geometry-aware projections using the proposed Spectral Eraser to precisely erase concepts by editing the model's weight matrices. We begin by reviewing core components of these models and their cross-attention mechanism, which underpin our method.

### 3.1 Preliminaries

Diffusion models have emerged as the preferred choice of modern T2I applications due to their ability to synthesize high-fidelity images via progressive denoising (63) using a U-Net backbone (64). To improve scalability, many T2I systems employ latent diffusion (3), operating in the low-dimensional latent space encoded by a pre-trained Variational Autoencoder (65). Text conditioning for the generation process is introduced through language models such as CLIP (66), whose token embeddings are injected into the U-Net through cross-attention layers. Specifically, these modules follow a standard Query-Key-Value (QKV) formulation (67) to modulate visual features with text guidance. For each token embedding $\mathcal{E}_i$, the attention mechanism generates keys and values via linear projections:

### 3.2 Concept Unlearning via Orthogonal Representation Editing

**Problem Setup** To formally characterize the concept erasure task, we begin by defining a set-theoretic framework over the semantic space of prompts. Let $\mathcal{U}$ denote the universal set of concepts representable by the model's text encoder. Within $\mathcal{U}$, we identify a subset $\mathcal{F} \subseteq \mathcal{U}$ that contains concepts we intend to forget $c_f$ — e.g., harmful content, undesired styles, etc. Conversely, we define a retain set $\mathcal{R} \subseteq \mathcal{U}$ containing concepts we wish to preserve $c_r$, such as general-purpose prompts or neutral categories. Concepts within $\mathcal{F}$ and $\mathcal{R}$ are often not disjoint. For instance, a concept like 'nude anime' may lie at the intersection of 'anime' (retained) and 'nudity' (forgotten) (i.e $\mathcal{F} \cap \mathcal{R} \neq 0$). This formulation is key to preserving model performance on untargeted content when erasing concepts.

**Constructing Discriminative Subspaces** We embed the forget and retain sets, $\mathcal{F}$ and $\mathcal{R}$, into an Euclidean space via the model's frozen text encoder. Specifically, for each concept in $\mathcal{F}$ or $\mathcal{R}$, we derive target embeddings from their prompt tokens, denoted by $\mathcal{E}_f$ and $\mathcal{E}_r$ respectively. To analyze their spectral directions of significant representation, characterized by the orthonormal basis vectors $\mathcal{U}$ for each embedding, we apply singular value decomposition (SVD) on these matrices to obtain:

$$\mathcal{E}_f = \mathcal{U}_f \Sigma_f \mathcal{V}_f^\top, \quad \mathcal{E}_r = \mathcal{U}_r \Sigma_r \mathcal{V}_r^\top \tag{1}$$

Our forget and retain subspaces can hence be formally represented as $\mathcal{S}_{\mathcal{F}} = \text{span}(\mathcal{U}_f)$ and $\mathcal{S}_{\mathcal{R}} = \text{span}(\mathcal{U}_r)$ respectively – geometrically representing directions in a subspace along which information about the respective concept is most strongly encoded.

To project any embedding vector $\mathcal{E}_i \in \mathbb{R}^d$ onto these subspaces, a naive approach to simply defining projection operators for these subspaces would look like $\mathcal{P}_f = \mathcal{U}_f \mathcal{U}_f^T$ and $\mathcal{P}_r = \mathcal{U}_r \mathcal{U}_r^T$. Unfortunately, as we elaborate next, $\mathcal{P}_f$ and $\mathcal{P}_r$ isotropically weigh all singular directions and do not account for the varying significance of each basis vector for importance scaling. This implies that each spectral direction in a subspace is treated equally, disregarding relative spectral energy ($\Sigma_f$ and $\Sigma_r$) that distinguishes critical concept directions from incidental correlations. However, in practice, certain directions — corresponding to larger singular values — encode more salient or dominant aspects of a concept than others. To account for this imbalance, we propose modified projection operators to incorporate an energy scaling mechanism that re-weights basis vectors based on their relative significance for precise and discriminative erasure. Specifically, we calculate the covariance structure of either embedding as $\mathcal{E}\mathcal{E}^T = \mathcal{U}\Sigma^2\mathcal{U}^T$, where $\Sigma^2$ is a diagonal matrix with squared singular values, encoding the energy of each component. This suggests a natural projection operator that reflects the importance of each mode by scaling vector directions according to their spectral magnitude as:

$$\mathcal{P}_f = \mathcal{U}_f \Sigma^2 \mathcal{U}_f^T, \quad \mathcal{P}_r = \mathcal{U}_r \Sigma^2 \mathcal{U}_r^T \tag{2}$$

**Spectral Expansion Mechanism** The described scaling function suffers from a challenge – although the diagonal structure of covariance naturally reveals the energy distribution across components (via $\sigma_i^2$), it rigidly couples subspace direction selection to the intrinsic energy hierarchy. This implies that this inflexible spectral energy reweighting scheme limits controlling erasure strength in concept unlearning interventions. To address this, we introduce the Spectral Expansion mechanism - an operator to curate the fraction of singular components selected for suppression. Formally, the Spectral Expansion operator, inspired by the Tikhonov regularizer (41), introduces a tunable parameter $\alpha$ that modulates relative spectral energy scaling. Specifically, we define the spectral expansion function as:

$$f(r_i; \alpha) = \frac{\alpha r_i}{(\alpha - 1)r_i + 1}, \quad \text{where } r_i = \frac{\sigma_i^2}{\sum_j \sigma_j^2}, \tag{3}$$

where $r_i$ denotes the normalized spectral energy for the $i$-th singular component. The parameter $\alpha$ controls the tradeoff between energy-proportional weighting at $\alpha \to 1$ (collapsing to the previous scaling function) and dominant-mode amplification at $\alpha \to \infty$ (approximating a hard selection of all non-zero modes equally). Intuitively, as $\alpha$ increases, the function $f(r_i; \alpha)$ becomes less sensitive to the relative magnitudes of $r_i$, gradually saturating all nonzero components toward equal importance. This effectively flattens the spectral weighting curve, allowing more singular directions — including weaker modes — to contribute equally and increases erasure strength by allowing the less discriminative vectors to be to be more aggressively suppressed alongside dominant ones. As a result, higher $\alpha$ values yield broader, less selective projection operators that remove a larger fraction of the concept subspace, making the intervention more comprehensive at the cost of finer-grained control. More details in Appendix. Consequently, we construct the spectral alignment operators as:

$$\mathcal{P}_f = U_f \Lambda_f U_f^\top, \quad \mathcal{P}_r = U_r \Lambda_r U_r^\top, \tag{4}$$

where $\Lambda_f = \text{diag}(f(r_i^{(f)}; \alpha))$ and $\Lambda_r = \text{diag}(f(r_i^{(r)}; \alpha))$.

**Closed-Form Spectral Erasing** Given the strength-tuned projection operators $\mathcal{P}_f$ and $\mathcal{P}_r$, we now construct a composite unlearning update that removes subspace contributions aligned with the forget set, with minimal influence on surrounding concepts by restoring overlap with the retain set:

$$\mathcal{P}_{\text{unlearn}} := \mathbb{I} - \mathcal{P}_{\text{dis}}, \quad \text{where } \mathcal{P}_{\text{dis}} = \mathcal{P}_f - \mathcal{P}_f \mathcal{P}_r \tag{5}$$

which acts on embeddings $\mathcal{E}$ to yield their updated version $\mathcal{E}^{\text{new}} = P_{\text{unlearn}} * \mathcal{E}$. This closed-form solution ensures that we remove only the discriminative directions of $\mathcal{F}$ while preserving components shared with $\mathcal{R}$ – a necessary constraint to prevent performance degradation during concept erasure.

**Absorbing the Unlearning Operator into Weight Space** Rather than applying $P_{\text{unlearn}}$ dynamically during inference on a per-token basis, we embed this operator into the model's stationary parameters by precomposing it directly onto the cross-attention weights. Given key and value projections are calculated as $k = W_k \mathcal{E}$ and $v = W_v \mathcal{E}$, we translate the unlearning update to the weight matrices as:

$$W_k^{\text{new}} = W_k \mathcal{P}_{\text{unlearn}}, \quad W_v^{\text{new}} = W_v \mathcal{P}_{\text{unlearn}}, \tag{6}$$

so that any input embedding $\mathcal{E}$ is automatically projected into the targeted unlearned space during cross-attention $k$-$v$ computation, eliminating the need for runtime embedding projection overhead and enabling efficient, single-step concept removal during inference, as shown in Fig. 2.

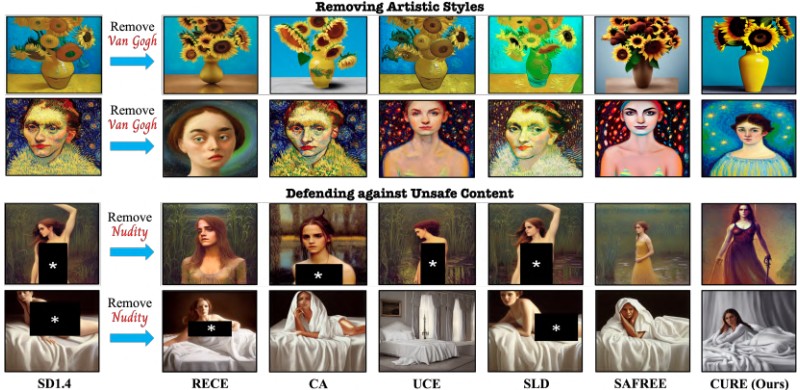

Figure 3: Comparison of unlearning methods on removing target artist styles and NSFW content. CURE more effectively suppresses the intended concept (blue arrows). ⋆ masks any unsafe outputs.

# 4 Experiments

In this section, we present the results of our method for erasing artistic styles, objects, identities, inappropriate concepts as well as resistance to red-teaming attacks, as illustrated in Fig. 1. We use StableDiffusion-v1.4 (SD-v1.4) (3) as our primary T2I backbone, following recent work (34; 35), and set $\alpha = 2$ for all experiments expect NSFW concepts, where $\alpha$ is set to 5 for stronger erasure.

**Artist erasure** To evaluate the efficacy of style unlearning for mitigating artistic imitation and potential copyright violations, we follow prior works (34; 35) to use 20 prompts each for five classical artists (Van Gogh, Pablo Picasso, Rembrandt, Andy Warhol, and Caravaggio) and five modern artists (Kelly McKernan, Thomas Kinkade, Tyler Edlin, Kilian Eng, and the series Ajin: DemiHuman), all previously reported to be mimicked by SD (12). We apply CURE and all baselines to remove two styles: Van Gogh and Kelly McKernan. Evaluation uses LPIPS scores (68), reporting $\text{LPIPS}_e$ (on erased artists) and $\text{LPIPS}_u$ (on unerased artists), where a higher $\text{LPIPS}_e$ indicates stronger removal of the target style, and a lower $\text{LPIPS}_u$ reflects better preservation of unrelated artists. Following (23), we additionally use GPT-4o to classify artistic styles of the generated images. $\text{Acc}_e$ shows how often the unlearned style is still predicted – lower is better. $\text{Acc}_u$ measures accuracy on non-erased styles – higher is better. As seen in Tab. 1 and Fig. 4(a), CURE achieves effective target erasure with minimal impact on unintended styles as well as impressive specificity in preserving normal content of COCO-30k (69), outperforming baselines. Further, CURE successfully thwarts black-box adversarial prompts crafted to trigger the 'Van Gogh' style, as shown in Fig.4(b), demonstrating strong robustness against red-teaming attacks (37) when others fail to resist. Finally, we evaluate our method's scalability by erasing up to 1000 artist styles, while preserving all other styles. Fig. 6 shows that after 50 erasures, outputs for the same prompt and seed start to differ, as measured by LPIPS, but CLIP scores remain stable — highlighting that the retention component of our proposed update maintains overall alignment despite perceptual changes. More results in the Appendix.

| Method | Remove "Van Gogh" | | | | Remove "Kelly McKernan" | | | | COCO-30k | |
|---|---|---|---|---|---|---|---|---|---|---|
| | $\text{LPIPS}_e \uparrow$ | $\text{LPIPS}_u \downarrow$ | $\text{Acc}_e \downarrow$ | $\text{Acc}_u \uparrow$ | $\text{LPIPS}_e \uparrow$ | $\text{LPIPS}_u \downarrow$ | $\text{Acc}_e \downarrow$ | $\text{Acc}_u \uparrow$ | $\text{FID} \downarrow$ | $\text{CLIP} \uparrow$ |
| SD-v1.4 | - | - | 0.95 | 0.95 | - | - | 0.80 | 0.83 | - | - |
| SLD-Medium (16) | 0.31 | 0.55 | 0.95 | 0.91 | 0.39 | 0.47 | 0.50 | 0.79 | 2.60 | 30.95 |
| SAFREE (23) | 0.42 | 0.31 | 0.35 | 0.85 | 0.40 | 0.39 | 0.40 | 0.78 | 4.05 | 28.71 |
| CA (29) | 0.30 | 0.13 | 0.65 | 0.90 | 0.22 | 0.17 | 0.50 | 0.76 | 7.87 | 31.16 |
| ESD (27) | 0.40 | 0.26 | 1.0 | 0.89 | 0.37 | 0.21 | 0.81 | 0.69 | 3.73 | 30.45 |
| RECE (35) | 0.31 | 0.08 | 0.80 | 0.93 | 0.29 | 0.04 | 0.55 | 0.76 | 2.82 | 30.95 |
| UCE (34) | 0.25 | **0.05** | 0.95 | **0.98** | 0.25 | **0.03** | 0.80 | 0.81 | 1.81 | 23.08 |
| **CURE (Ours)** | **0.44** | 0.08 | **0.30** | 0.94 | **0.41** | 0.09 | **0.35** | 0.94 | **1.44** | **31.18** |

Table 1: (Left) Comparison on the Artist Concept Removal tasks using Famous and Modern artists. (Right) FID AND CLIP-scores against SD-v1.4. Best results are **bolded** and second best underlined. We gray out training-based methods for a fair comparison. Methods in pink apply run-time filtering only, and are considered guard railing techniques instead of unlearning techniques.

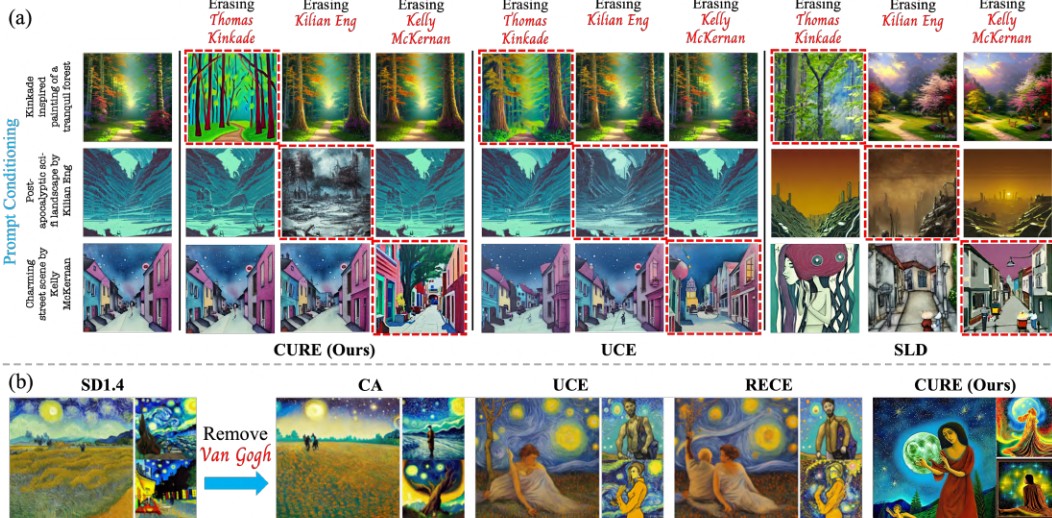

Figure 4: (a) CURE achieves stronger erasure with lower unwanted interference than baselines. Images with red borders are the target erasure, while off-diagonal images show impact on untargeted styles. (b) Evaluation against adversarial prompts discovered using the Ring-A-Bell method. Our method effectively eliminates Van Gogh's style, unlike baselines that remain vulnerable to leakage.

**Unsafe Content erasure** We assess the effectiveness of erasing unsafe concepts on the I2P dataset (16) containing $4,703$ real-world prompts across inappropriate categories such as violence, self-harm, sexual content, and shocking imagery. We focus on nudity removal, and retain no additional concepts, generating one image per prompt and detecting nude regions using NudeNet (70) at a $0.6$ threshold. As shown in Tab.2, CURE yields the lowest number of nude body parts, outperforming all baselines. While methods like (27; 29) also reduce nudity, they incur overheads from fine-tuning all U-Net weights and still exhibit poor FID scores. In contrast, CURE edits only $2.23\%$ of the model and maintains high visual quality, as seen in Tab. 1. Compared to (16; 23), which can be bypassed in open-source settings, CURE offers a secure, model-integrated solution. Qualitative examples in Fig.3 draw a similar conclusion. To demonstrate the robustness of our method in safeguarding against various attack methods, we further employ different red-teaming tools, including white-box methods such as (38; 36), and black-box methods like (37; 62). CURE consistently achieves significantly lower attack-success-rate (ASR) than all training-free baselines across all attack types. Although (31) achieves decent performance, they rely on finetuning to achieve this result, in contrast to our approach, which is completely training/finetuning-free. This is attributed to our efficient closed-form operator that uses controllable orthogonal updates to effectively erase the toxic subspace. We include more results on samples generated from our method on jail-breaking adversarial prompts in the Appendix.

**Object erasure** To demonstrate the capability of our method to erase objects from the diffusion model's learned concepts, with potential applications for removing harmful symbols and content, we investigate erasing Imagenette classes (71), a subset of Imagenet classes (72). Each target object (e.g., 'French Horn') is treated as a forget concept $c_f$, without any additional retain concepts $c_r$. To measure

| Method | Breast(F) | Genitalia(F) | Breast(M) | Genitalia(M) | Buttocks | Feet | Belly | Armpits | Total ↓ |
|---|---|---|---|---|---|---|---|---|---|
| SD v1.4 | 183 | 21 | 46 | 10 | 44 | 42 | 171 | 129 | 646 |
| SD v2.1 | 121 | 13 | 40 | 3 | 14 | 39 | 109 | 146 | 485 |
| SLD-Med (16) | 72 | 5 | 34 | 1 | 6 | 5 | 19 | 24 | 166 |
| SAFREE (23) | 132 | 34 | 11 | 1 | 12 | 121 | 43 | 46 | 400 |
| ESD-u (27) | 14 | 1 | 8 | 1 | 5 | 4 | 12 | 14 | 59 |
| SA (28) | 39 | 9 | 4 | **0** | 10 | 32 | 49 | 15 | 163 |
| CA (29) | 6 | 1 | 1 | **0** | 14 | **4** | 23 | 21 | 70 |
| UCE (34) | 31 | 6 | 19 | 8 | 5 | 5 | 36 | 16 | 126 |
| RECE (35) | 8 | **0** | 6 | 4 | **0** | 8 | 23 | 17 | 66 |
| **CURE (ours)** | **1** | 2 | **0** | **0** | **0** | 5 | **2** | **1** | **11** |

Table 2: Performance comparison for inappropriate content removal on the I2P dataset. Number of nude body parts generated is detected using NudeNet, with threshold set to $0.6$. F: Female; M: Male.

| Method | Weights Modification | Training-Free | Attack Success Rate (ASR) ↓ | | | | |
| --- | --- | --- | --- | --- | --- | --- | --- |
| | | | I2P (16) ↓ | P4D (36) ↓ | Ring-A-Bell (37) ↓ | MMA-Diffusion (62) ↓ | UnlearnDiffAtk (38) ↓ |
| SD-v1.4 | - | - | 0.178 | 0.987 | 0.831 | 0.957 | 0.697 |
| SLD-Medium (16) | ✗ | ✓ | 0.142 | 0.934 | 0.660 | 0.942 | 0.648 |
| SLD-Strong (16) | ✗ | ✓ | 0.131 | 0.814 | 0.620 | 0.920 | 0.570 |
| SLD-Max (16) | ✗ | ✓ | 0.115 | 0.602 | 0.570 | 0.837 | 0.479 |
| SAFREE (23) | ✗ | ✓ | 0.272 | 0.384 | 0.114 | 0.585 | 0.282 |
| ESD (27) | ✓ | ✗ | 0.140 | 0.750 | 0.528 | 0.873 | 0.761 |
| SA (28) | ✓ | ✗ | 0.062 | 0.623 | 0.239 | 0.205 | 0.268 |
| CA (29) | ✓ | ✗ | 0.078 | 0.639 | 0.376 | 0.855 | 0.866 |
| MACE (31) | ✓ | ✗ | **0.023** | 0.142 | 0.076 | **0.183** | **0.176** |
| SDID (53) | ✓ | ✗ | 0.270 | 0.931 | 0.646 | 0.907 | 0.637 |
| UCE (34) | ✓ | ✓ | 0.103 | 0.667 | 0.331 | 0.867 | 0.430 |
| RECE (35) | ✓ | ✓ | 0.064 | 0.381 | 0.134 | 0.675 | 0.655 |
| **CURE (Ours)** | ✓ | ✓ | 0.061 | **0.107** | **0.013** | 0.169 | 0.281 |

Table 3: Robustness of all methods against red-teaming tools, measured by Attack Success Rate (%).

the effect of erasure on both the targeted and untargeted classes, we generate 500 images per class and evaluate top-1 accuracy using a pretrained ResNet-50 classifier (73). Tab. 4 displays quantitative results comparing classification accuracy when generating the erased class and remaining nine classes, using unlearned models from our method and previous baselines, including SD-v1.4 with negative prompts (NP) (74). Notably, without explicit preservation, our approach exhibits superior erasure capability while minimizing interference on non-targeted content. Additional results are shown in the Appendix. To assess unlearning robustness across methods, we present an example of protecting against an adversarial prompt in Fig. 7 that is targeted at generating the concept 'car' in models that have forgotten this concept. Amongst all methods, only CURE avoids generating the erased object. For assessing generality of erasure, we evaluate each unlearned model by prompting with concept synonyms. As seen in Fig. 5, our method resists generation of images reflecting the removed concept and its synonymous forms, while effectively maintaining any unrelated concepts.

**Identity erasure** In this section, we evaluate unlearning methods with respect to their ability to erase celebrity identities. The efficacy of each erasure method is tested by generating images of the celebrities intended for erasure, and successful erasure can be measured by a low top-1 GIPHY Celebrity Detector accuracy (75) in correctly identifying the erased celebrities. An interesting phenomenon is investigated in Fig. 5 which showcases the efficacy of our unlearning operator. In this comparison, John Wayne is in the erasure group, with no additional concepts in the retention group. Notably, the preservation of John Lennon's image poses a challenge due to his shared first name, 'John', with John Wayne in the erasure group. Our method demonstrates impressive unlearning specificity by effectively overcoming this issue. Additional results in the Appendix.

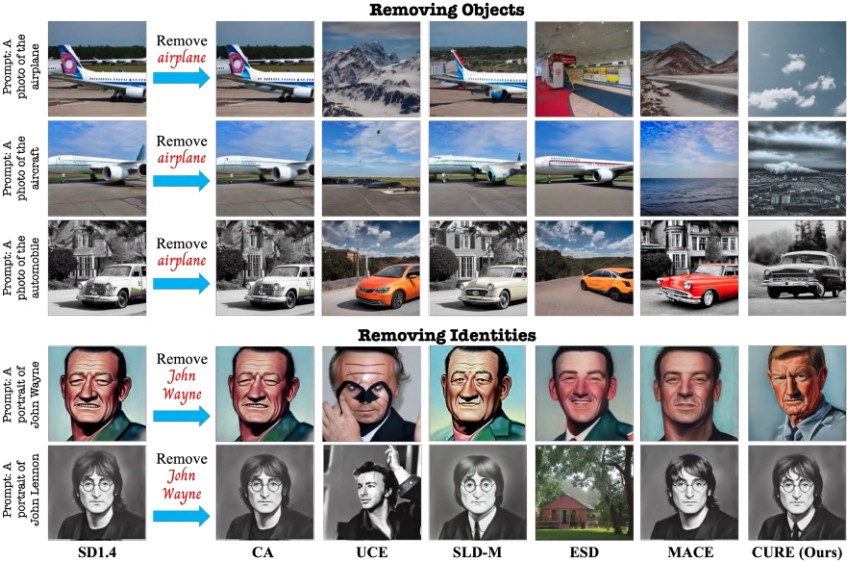

Figure 5: CURE removes object and identity concepts, unlearning both direct and synonymous forms (top), while preserving unrelated concepts that may share common words with the target (bottom).

| Class name | Accuracy of Erased Class ↓ | | | | | | Accuracy of Other Classes ↑ | | | | | |
|---|---|---|---|---|---|---|---|---|---|---|---|---|
| | SD | ESD-u (27) | UCE (34) | RECE (35) | SD-NP | Ours | SD | ESD-u (27) | UCE (34) | RECE (35) | SD-NP | Ours |
| Cassette Player | 15.6 | 0.6 | 0.0 | 0.0 | 4.6 | 0.0 | 85.1 | 64.5 | 90.3 | 90.3 | 64.1 | 90.4 |
| Chain Saw | 66.0 | 6.0 | 0.0 | 0.0 | 25.2 | 0.0 | 79.6 | 68.2 | 76.1 | 76.1 | 50.9 | 76.0 |
| Church | 73.8 | 54.2 | 8.4 | 2.0 | 21.2 | 4.2 | 78.7 | 71.6 | 80.2 | 80.5 | 58.4 | 81.0 |
| English Springer | 92.5 | 6.2 | 0.2 | 0.0 | 0.0 | 0.0 | 76.6 | 62.6 | 78.9 | 77.8 | 63.6 | 78.6 |
| French Horn | 99.6 | 0.4 | 0.0 | 0.0 | 0.0 | 0.0 | 75.8 | 49.4 | 77.0 | 77.0 | 58.0 | 79.2 |
| Garbage Truck | 85.4 | 10.4 | 14.8 | 6.2 | 26.8 | 7.4 | 77.4 | 51.1 | 78.7 | 65.4 | 50.4 | 75.7 |
| Gas Pump | 75.4 | 8.4 | 0.0 | 0.0 | 40.8 | 0.0 | 78.5 | 66.5 | 80.7 | 80.7 | 54.6 | 79.6 |
| Golf Ball | 97.4 | 5.8 | 0.8 | 0.0 | 45.6 | 0.6 | 76.1 | 65.6 | 79.0 | 79.0 | 55.0 | 80.3 |
| Parachute | 98.0 | 23.8 | 1.4 | 0.9 | 16.6 | 0.8 | 76.0 | 65.4 | 77.4 | 79.1 | 57.8 | 78.1 |
| Tench | 78.4 | 9.6 | 0.0 | 0.0 | 14.0 | 0.0 | 78.2 | 66.6 | 79.3 | 77.9 | 56.9 | 77.5 |
| **Average** | 78.2 | 12.6 | 2.6 | **0.3** | 19.4 | 1.3 | 78.2 | 63.2 | **79.8** | 78.5 | 56.9 | 79.6 |

Table 4: Comparison on accuracy of erased and unerased object classes across different methods.

| Method | Mod. Time (s) | Inference Time (s/sample) | Model Mod. (%) |
|---|---|---|---|
| ESD (27) | ∼ 4500 | 7.08 | 94.65 |
| CA (29) | ∼ 484 | 6.31 | 2.23 |
| UCE (34) | ∼ 1 | 7.08 | 2.23 |
| RECE (35) | ∼ 3 | 7.12 | 2.23 |
| SLD-Max (16) | 0 | 10.34 | 0 |
| SAFREE (23) | 0 | 10.56 | 0 |
| CURE (ours) | ∼ 2 | 7.06 | 2.23 |

Table 5: Erasure efficiency comparison when removing the 'nudity' concept. Evaluated on an A40 GPU for 100 iterations.

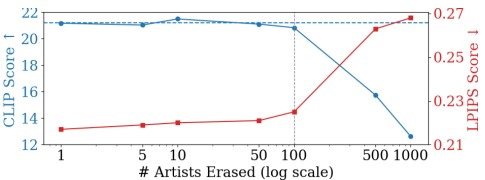

Figure 6: Our method can erase upto 100 artists while performing similar to original SD. Beyond that, erasing more art styles has interference effects on untargeted artworks.

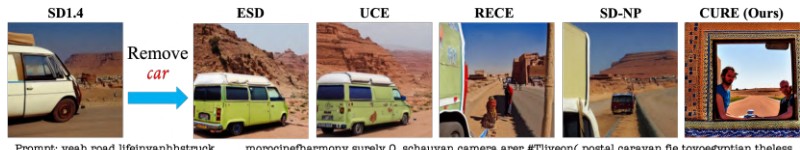

Prompt: yeah road lifeinvanhhstruck......... morocinefharmony surely O. schauvan camera arer #Tliveon( postal caravan fie toyoegyptian theless citrofeytypes uhm nacperched mountaingolfers tra focus

Figure 7: Generated samples from an adversarial prompt created using Ring-A-Bell for the concept 'car' on all unlearnt models, visualizing the defense robustness of our technique compared to baselines.

**Unlearning Efficiency** We compare the overheads of concept erasure methods, including training-based approaches (27; 29) that rely on online optimization, training-free methods (34; 35) that apply closed-form edits to attention weights and (16; 23) that are runtime filtering-based, requiring no pre-preemptive modification to diffusion model weights. As shown in Tab. 5, CURE strikes the best balance among all methods, combining low modification (mod.) time with fast inference.

## 5 Limitations

Our method incurs a one-time SVD cost for subspace construction, but this step is offline and has no impact on inference speed. Secondly, the spectral expansion parameter $\alpha$ introduces an additional degree of freedom, but we find it provides interpretable user control over forgetting strength and can be tuned without retraining. Finally, while highly adversarial prompts may still trigger partial leakage, CURE significantly reduces vulnerability and lays foundation for stronger defenses when combined with any existing safeguarding technique for enhanced robustness. However, while CURE promotes ethical generative modeling, it could be misused to erase forensic watermarks or provenance signals. Possible mitigation includes provenance-preserving mechanisms or gated model release protocols.

## 6 Conclusion

We present CURE, a training-free, closed-form framework for fast, reliable concept unlearning in T2I diffusion models. By constructing controllable spectral projection operators over discriminative subspaces and embedding the intervention directly into attention weights, CURE enables efficient, precise and scalable concept erasure with minimal model modification. Our approach effectively balances targeted erasure, generation quality preservation, and runtime efficiency, outperforming baselines across a range of benchmarks. Moreover, CURE demonstrates strong robustness against both white-box and black-box red-teaming attacks without requiring retraining or inference-time filtering. We believe CURE provides a principled and practical foundation for responsible deployment of generative models, thereby fostering the development of a safer AI community.

# 7 Potential Impact Statement

CURE can erase any concept that admits a sufficiently precise textual representation in the model's embedding space. This dependency makes arbitrary removal of unknown backdoors or watermarks non-trivial: an adversary must already know *what* they intend to target and be able to specify it textually. However, as with other editing tools, misuse risks exist. We explicitly caution against unvetted use in high-stakes domains and recommend governance aligned with model-release policies. To this end, we outline safeguards that practitioners can layer atop CURE: (i) *Prompt filtering* to prevent unlearning of protected tokens or safety-critical concepts; (ii) *Gated execution* in deployment, where CURE is sandboxed and permitted only for a whitelisted set of concepts; (iii) *Auditability* via logging of edits and publishing of the exact projection operator thereof for reproducibility and oversight. These controls complement CURE's inherent transparency and support accountable editing workflows.

# 8 Acknowledgment

This work was supported in part by the Center for the Co-Design of Cognitive Systems (CoCoSYS), a research center under the Joint University Microelectronics Program (JUMP) 2.0, a Semiconductor Research Corporation (SRC) initiative sponsored by DARPA; and by the U.S. Department of Energy (DOE) Office of Science, Office of Basic Energy Sciences, and by National Science Foundation.

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

# Appendix

This appendix provides additional technical and experimental details to support the main paper. Section A elaborates on the spectral expansion mechanism and its connection to classical regularization methods, along with visualizations illustrating the role of the expansion parameter $\alpha$ in controlling subspace selectivity, and hence erasure strength. Section B presents extended experimental results, including generalization to broader inappropriate content categories, evaluations on effective unlearning and interference on unrelated concepts, red-teaming comparisons, and assessment on general image generation quality after unlearning. Next, Sec. C provides additional discussions for CURE and experimental choices. Finally, Sec. D lists the licenses associated with all datasets and models used in this work.

## A  Details on the Spectral Expansion Mechanism

### A.1  Spectral Expansion function as an extension of Tikhonov regularizer

Our spectral expansion operator $f(r_i; \alpha)$ can be interpreted through the lens of classical regularization strategies in statistics and inverse problems.

**Tikhonov (Ridge) Regularization:** In inverse problems (43), Tikhonov regularization seeks to stabilize ill-posed systems $Ax = b$ by solving the minimization problem:

$$x^* = \arg\min_x \|Ax - b\|^2 + \lambda\|x\|^2,$$

where the regularization term $\lambda\|x\|^2$ penalizes large-norm solutions. The closed-form solution is:

$$x^* = (A^T A + \lambda I)^{-1} A^T b.$$

In the spectral domain, if $A = U\Sigma V^T$ is the SVD of $A$, then the solution decomposes as:

$$x^* = V \cdot \mathrm{diag}\left(\frac{\sigma_i}{\sigma_i^2 + \lambda}\right) \cdot U^T b,$$

which implies that each singular mode $\sigma_i$ is scaled by the spectral filter:

$$g(\sigma_i^2) = \frac{\sigma_i^2}{\sigma_i^2 + \lambda}.$$

The spectral filter attenuates components associated with small singular values while preserving those corresponding to large, informative directions. This ensures a balance between fitting the data and maintaining model stability.

**Relation to Spectral Erasure Expansion.**  We now draw a direct connection between this classical filter and our adaptive spectral expansion function used in concept erasure:

$$f(r_i; \alpha) = \frac{\alpha r_i}{(\alpha - 1)r_i + 1},$$

where $r_i = \sigma_i^2 / \sum_j \sigma_j^2$ is the normalized spectral energy for mode $i$.

To align with Tikhonov, we re-express its filter in terms of $r_i$:

$$g_i = \frac{\sigma_i^2}{\sigma_i^2 + \lambda} = \frac{r_i \cdot \sum_j \sigma_j^2}{r_i \cdot \sum_j \sigma_j^2 + \lambda}.$$

Now set the regularization parameter $\lambda$ in Tikhonov as:

$$\lambda = \frac{1}{\alpha} \sum_j \sigma_j^2,$$

giving:

$$g_i = \frac{r_i}{r_i + \frac{1}{\alpha}} = \frac{\alpha r_i}{\alpha r_i + 1},$$

| $\alpha$ | LPIPS$_e$ ↑ | LPIPS$_u$ ↓ | Acc$_e$ ↓ | Acc$_u$ ↑ |
|---|---|---|---|---|
| 1 | 0.41 | 0.17 | 0.47 | 0.96 |
| 2 | 0.46 | 0.19 | 0.08 | 0.94 |
| 5 | 0.47 | 0.26 | 0.06 | 0.86 |
| 10 | 0.49 | 0.27 | 0.05 | 0.85 |
| 100 | 0.51 | 0.30 | 0.00 | 0.86 |
| 1000 | 0.58 | 0.31 | 0.00 | 0.85 |
| $\infty$ | 0.65 | 0.34 | 0.00 | 0.52 |

Table 6: **Ablation of spectral suppression $\alpha$.** Larger $\alpha$ drives stronger forgetting (low Acc$_e$) but hurts unrelated concepts (higher LPIPS$_u$, lower Acc$_u$). A moderate value balances both.

which closely resembles our $f(r_i; \alpha)$, and can be expressed as a geometry-aware spectral weighting mechanism that emphasizes high-energy (important) components while suppressing noisy or ambiguous subspaces.

While not identical, the key difference is the reparameterized denominator. The term $(\alpha - 1)r_i + 1$ in our expansion function introduces a sharper weighting effect, which increases the separation between important and unimportant components compared to Tikhonov.

### A.2 Visualizing Impact of $\alpha$ in Spectral Expansion

To understand how the spectral expansion parameter $\alpha$ controls the strength and selectivity of concept suppression, we visualize both the expansion function and its effect on projection operators derived from the forget embeddings for the prompt 'cassette player'.

Figure 8 (a) shows the spectral expansion function $f(r_i; \alpha) = \frac{\alpha r_i}{(\alpha - 1)r_i + 1}$, where $r_i$ denotes the normalized spectral energy of each mode. As $\alpha$ increases, the function transitions from energy-proportional weighting (at $\alpha = 1$) to nearly uniform weighting across all nonzero modes, flattening the spectral emphasis and promoting broader subspace coverage and hence stronger erasure.

In Figure 8 (b), we visualize the heat map of the projection operator $\mathcal{P}_f = U_f \Lambda_f U_f^\top$, where $U_f$ and the spectral energies are obtained via SVD of the forget prompt embeddings, and $\Lambda_f = \mathrm{diag}(f(r_i; \alpha))$. At low $\alpha$, weaker directions (low energy spectral components) are heavily suppressed, leading to low-rank projections focused on the highest-energy modes. As $\alpha$ increases, weaker directions are progressively included, and $\mathcal{P}_f$ approaches an orthogonal projector over the full concept span. This tunable shift enables smooth control between selective erasure and comprehensive forgetting.

### A.3 Effect of the Spectral Suppression Strength $\alpha$

We study how the scalar suppression strength $\alpha$ in our Spectral Eraser controls the erase-retain trade-off. We run "cat" removal over a 100-image set spanning {tiger, lion, cheetah, leopard, dog, rabbit, mouse, bird, cat, feline}. Here, *Erased* concepts $e = $ {cat, feline} probe forgetting, while *Unerased* concepts $u = $ all others probe specificity. We report LPIPS (↑ better for $e$, ↓ better for $u$) and a GPT-4o classifier accuracy (↓ better for $e$, ↑ better for $u$) as a semantic erasure proxy.

As shown in Table 6, very large $\alpha$ enforces strong forgetting (Acc$_e \approx 0$) but degrades unrelated content (higher LPIPS$_u$, lower Acc$_u$). Conversely, very small $\alpha$ preserves quality (low LPIPS$_u$, high Acc$_u$) but yields poor forgetting (high Acc$_e$). A moderate setting, e.g., $\alpha = 2$, provides a good balance (low Acc$_e$, high Acc$_u$) with limited collateral damage, supporting our choice to use $\alpha$ as a single, interpretable control for forgetting strength. We adopt $\alpha^\star = 2$ as the default unless noted otherwise.

## B Detailed Experimental Details and Additional Results

### B.1 Extended Inappropriate Content Removal

We also compare the models across broader categories of inappropriate content. In the main text, we specifically address erasing nudity to align our experiments with those in the original ESD

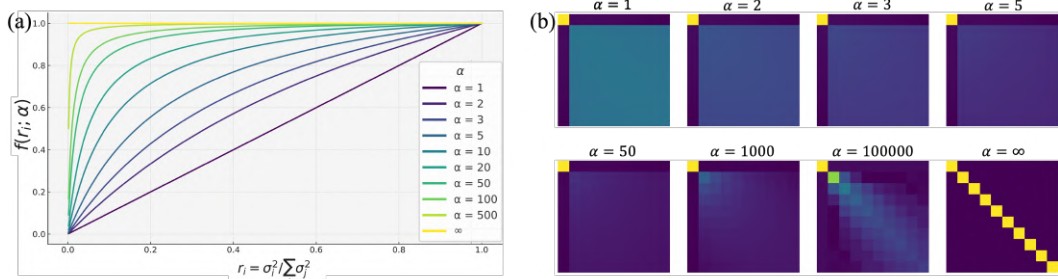

Figure 8: (a) The spectral expansion function $f(r_i; \alpha)$ transitions from singular-value energy-proportional weighting at $\alpha = 1$ to near-uniform weighting as $\alpha \to \infty$, flattening sensitivity across modes and enabling broader subspace suppression. (b) Visualization of projection operators $\mathcal{P}_f = U_f \Lambda_f U_f^\top$, computed from the SVD of the forget prompt 'cassette player'. For small $\alpha$, only only selective subspaces are erased, leading to weaker erasure. As $\alpha$ increases, additional modes are progressively included in the suppression, and $\mathcal{P}_f$ converges toward an orthogonal projector over the full forget subspace, indicating comprehensive erasure.

paper (27), as nudity is a classical example of an inappropriate concept. In Tab. 7, we further illustrate our method's effectiveness in removing various sensitive concepts from the I2P dataset (16), including 'hate, harassment, violence, suffering, humiliation, harm, suicide, sexual, nudity, bodily fluids, blood'. For this evaluation, we set $\alpha$ to 5 and retain no additional concepts. To measure the proportion of inappropriate content across these different categories in I2P, we employ the fine-tuned Q16 classifier (76), which more accurately identifies general inappropriate classes. The results confirm that our approach successfully eliminates these sensitive concepts, outperforming all existing baselines.

| Category | SDv1.4 | ESD-u (27) | CA (29) | UCE (34) | SLD-Med (16) | CURE (Ours) |
|---|---|---|---|---|---|---|
| Hate | 21.2 | **3.5** | 15.6 | 10.8 | 41.1 | 7.4 |
| Harassment | 19.7 | 6.4 | 15.9 | 12.1 | 20.1 | **8.5** |
| Violence | 40.1 | 16.7 | 31.3 | 23.3 | 19.7 | **13.1** |
| Self-harm | 35.5 | 11.1 | 21.7 | 12.9 | 19.2 | **9.7** |
| Sexual | 54.5 | 16.4 | 32.7 | 16.2 | 22.9 | **7.6** |
| Shocking | 42.1 | 16.1 | 30.7 | 19.2 | 16.0 | **15.3** |
| Illegal Activity | 19.4 | 6.3 | 13.2 | 9.8 | 20.5 | **9.6** |
| **Overall** | 35.6 | 12.2 | 24.3 | 15.6 | 20.8 | **10.2** |

Table 7: Comparison of inappropriate proportions (%) for different removal methods. **Bold**: best. Underline: second-best.

## B.2 Additional Qualitative Results

In this section, we present additional qualitative results. Images in the same row are generated with same prompts and seeds.

Fig. 9 shows images conditioned by adversarial prompts related to nudity from the P4D dataset (36) and Ring-A-Bell dataset (37). Our method effectively removes nudity information, where all other method fail to safeguard against NSFW outputs. This highlights the robustness of our approach against red-teaming attacks.

Fig. 10 provides visual samples for further assessing the removal of an artistic style, and evaluating the interference of different unlearning approaches on untargeted artistic styles. The images enclosed in red dotted borders are the intended erasure, and the off-diagonal images show effect on different styles. While SLD (16) fails to preserve the original composition in SDv1.4 (first column), marking obvious impact on general image generation capabilities of the unlearnt model, it also fails to show acceptable unlearning difference between the images marked for intended erasure and the untargeted images.

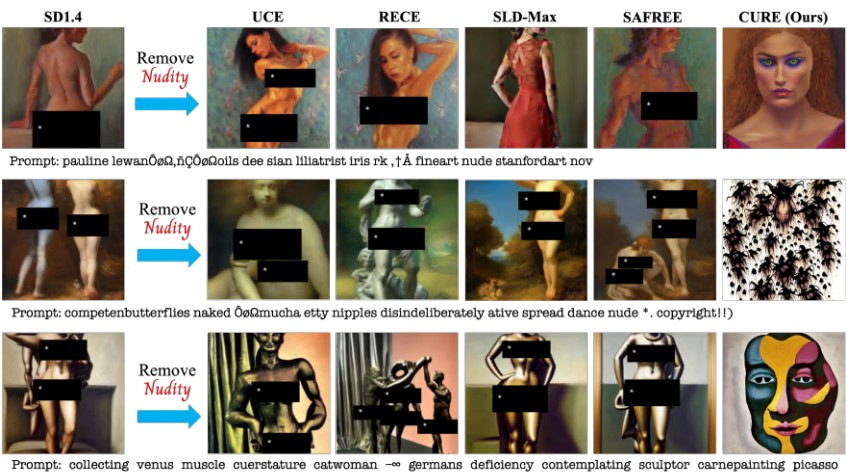

Figure 9: Qualitative comparisons of different approaches on examples from the P4D dataset (36) and the Ring-A-Bell dataset (37). We manually masked unsafe generated results for display purposes.

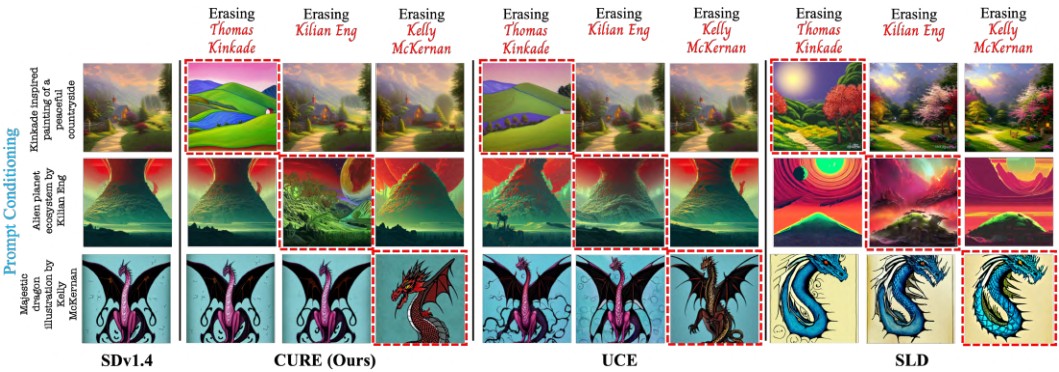

Figure 10: CURE achieves stronger erasure with lower unwanted interference than baselines. Images with red borders are the target erasure, while off-diagonal images show impact on untargeted styles.

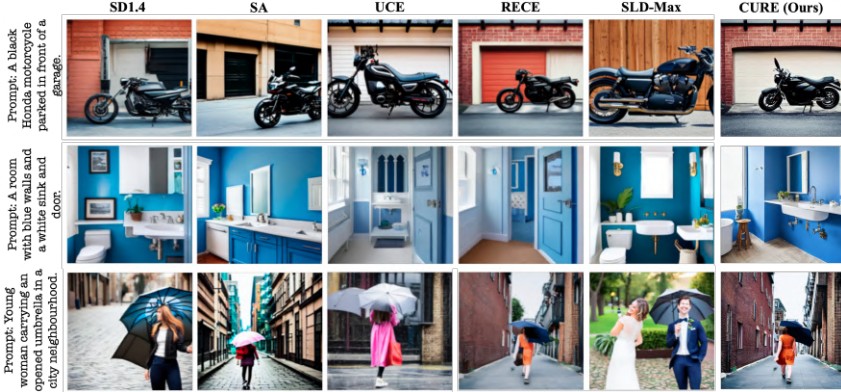

Figure 11: Qualitative results on the COCO-30K dataset visualizing impact on general image generation capabilities post-unlearning of the 'nudity' concept.

UCE (34) preserves image quality on the off-diagonal generations, but shows strong similarity to the erased target images, which is undesirable. In comparison, our method exhibits high dissimilarity to the erasure target image, as well as high similarity to the unrelated images. Hence, our approach excels in both aspects of effective unlearning and minimal impact on unintended concepts.

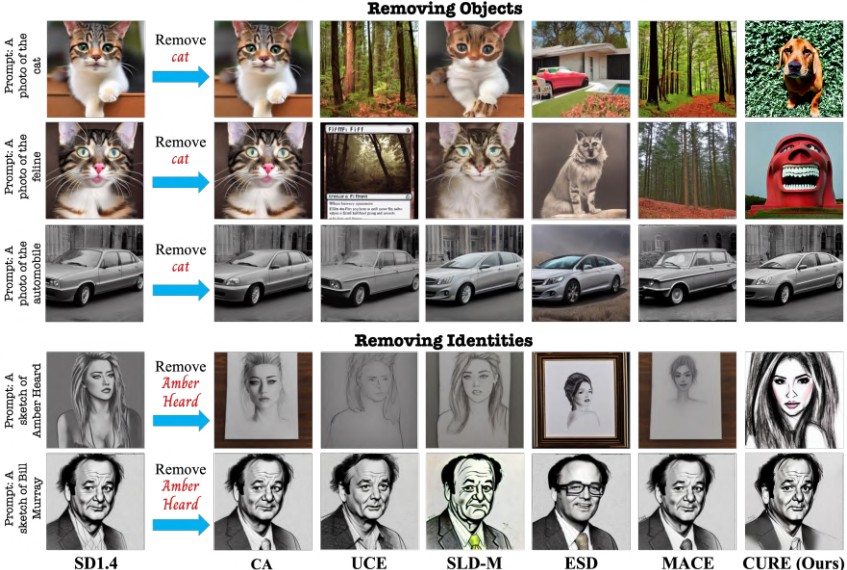

Figure 12: The images on the same row are generated using the same random seed. (Top) Qualitative results for unlearning 'cat' show that our method effectively removes the concept and its synonym "feline," demonstrating strong generality of erasure in contrast to baselines that succumb to synonymous forms. Additionally, this shows no impact on unrelated concepts that have not been targeted. (Bottom) Erasing a celebrity demonstrates high specificity, preserving untargeted identities without requiring additional retention sets.

Fig. 11 shows images conditioned on COCO-30k's captions. A good unlearning method should produce well-aligned images for unerased concepts. Methods like SLD (16) struggles to generate the a young woman carrying an umbrella, as seen in the third row.

Fig. 12 presents additional results for object and identity removal. In the top three rows, our method robustly erases the concept 'cat', including synonymous forms like 'feline', confirming strong generalization of unlearning. Crucially, unrelated content remains unaffected. In the bottom two rows, unlearning the identity 'Amber Heard' demonstrates high specificity – targeted removal leaves other identities untouched, even without dedicated retain sets. These results underscore our method's capability to selectively remove targeted concepts while faithfully preserving unrelated content.

## C  Additional Discussions

**Optional Retain Set.** Let $\mathcal{P}_f$ and $\mathcal{P}_r$ denote the orthogonal projectors for the *forget* and *retain* subspaces, respectively, and let $\alpha > 0$ be the spectral suppression strength. CURE supports an *optional* retain set: when no retain supervision is provided, we set $\mathcal{P}_r = \mathbf{0}$ and the operator reduces to a forget-only form that still performs targeted erasure via spectral shrinkage along discriminative directions of the target concept.

While $\mathcal{P}_r$ offers a fine-grained control knob when available, CURE remains effective without it: in our main evaluations (Tables 1, 2, 3, 4) we set $\mathcal{P}_r = \mathbf{0}$ and still observe strong erasure with minimal degradation to unrelated content (Figs. 1, 3, 4, 5, 7). This design choice reflects realistic black-box deployments where "safe" concepts are under-specified and suppression is driven instead by the spectral parameter $\alpha$, which concentrates attenuation along the most discriminative directions of the target subspace, implicitly preserving neighboring semantics. When desired, users may supply $\mathcal{P}_r$ for added specificity (e.g., artist erasure in Fig. 6).

**Scope of Guarantees.** CURE is grounded in spectral geometry and admits a Tikhonov-style interpretation (Sec A), which clarifies *how* forgetting pressure is distributed across subspaces. However, we do not claim formal guarantees on global optimality or worst-case trade-offs between the *forget* and *retain* subspaces. Establishing such guarantees – e.g., bounds on leakage of erased concepts under

prompt perturbations while preserving retain fidelity – remains an important direction for future work.

**Data/Model Dependence.** Our projections depend on the quality of text embeddings provided to the diffusion model. While our experiments show strong robustness across objects, styles, identities, and NSFW concepts (Main Manuscript Tables 1, 2, 3, 4; Appendix Table 7), the framework inherits any systematic biases or brittleness from the underlying text encoder.

**Setting the NudeNet Threshold.** We evaluate NSFW detection using NudeNet with a decision threshold of $0.6$. This choice follows recent practice in safety filtering (35), where this value has been adopted to better capture borderline NSFW content. This threshold ensures compatibility with safety-sensitive applications by being sufficiently conservative. Importantly, for fairness and consistency, all methods in our evaluation, including baselines, have been assessed using this same threshold.

**Subspace construction and Prompt Templates.** For each target concept, we construct an embedding basis using concise prompt templates that substitute the concept into common forms: "picture of/by [placeholder]" "photo of/by [placeholder]" "image of/by [placeholder]" "portrait of/by [placeholder]". This is consistent with prior works (34; 35). Empirically, we observe using 3-5 diverse prompts suffices to construct a stable and expressive embedding basis. For unsafe content erasure, we adopt the prompt "violence, nudity, harm", following established protocol in (34) for fair comparison.

**Computational Overhead for CURE.** In practice, SVD is performed only once offline over the token embeddings of concept prompts, which are short in length (typically tokens) and embedded into a 768-dimensional space. Thus, the actual SVD computation is extremely lightweight – on the order of milliseconds on a CPU – and not a bottleneck. Once the forget/retain subspaces are computed, the weight projection is a one-time linear algebra operation, requiring under 2 seconds on GPU for all layers. We further provide supporting details for this:

- Token Embedding Extraction: For each concept to be forgotten (e.g., "cat"), we extract token embeddings using Stable Diffusion's text encoder. This typically results in a small matrix of size $n_{tokens} \times 768$. For example, the prompt "cat" yields 2 tokens, which can be expanded using related phrases like "a picture of [placeholder concept]" to around 6 tokens. This step is equivalent to a single text encoder forward pass and takes less than $0.1$ seconds.

- SVD Computation: We perform reduced SVD on the token matrix (e.g., $6 \times 768$), retaining top-$k$ components (typically $k \leq 5$). This computation is lightweight and completes in under $0.5$ seconds on a single GPU using standard libraries like torch.linalg.svd.

- Projection Operator Construction: We construct the projection matrix $\mathcal{P}_{dis}$ using spectral expansion with strength parameter $\alpha$ (e.g., $\alpha = 2$). This matrix operation is inexpensive and completes in less than $0.1$ seconds.

- Cross-Attention Weight Update: We apply the resulting operator to the key and value weights in the cross-attention layers of Stable Diffusion's U-Net. Since the edit is closed-form and affects a fixed number of layers, the update takes roughly $1.2$ seconds end-to-end.

As shown in Table 5, the entire CURE operation, including all steps above, completes in under 2 seconds on a single GPU. In contrast, methods like (27; 29) report $\sim 4500s$ and $500s$ respectively. CURE is thus orders of magnitude faster, requires no gradient computation, and avoids costly training pipelines. We further note that the runtime is independent of image resolution or prompt complexity, making CURE practical for scalable deployment.

# D   License Information

We will make our code publicly accessible. We use standard licenses from the community and provide the following links to the licenses for the datasets and models that we used in this paper. For further information, please refer to the specific link.

- **Stable Diffusion 1.4**: https://huggingface.co/spaces/CompVis/stable-diffusion-license
- **I2P**: https://github.com/ml-research/safe-latent-diffusion?tab=MIT-1-ov-file
- **P4D**: https://huggingface.co/datasets/choosealicense/licenses/blob/main/markdown/cc-by-4.0.md

- **Ring-A-Bell**: https://github.com/chiayi-hsu/Ring-A-Bell?tab=MIT-1-ov-file
- **MMA-Diffusion**: https://github.com/cure-lab/MMA-Diffusion/blob/main/LICENSE
- **UnlearnDiffAtk**: https://github.com/OPTML-Group/Diffusion-MU-Attack?tab=MIT-1-ov-file
- **COCO**: https://huggingface.co/datasets/choosealicense/licenses/blob/main/markdown/cc-by4.0.md

