# OpenReview forum: "CURE: Concept Unlearning via Orthogonal Representation Editing in Diffusion Models"
_NeurIPS.cc/2025/Conference — NeurIPS 2025 spotlight_

### Official Review · Reviewer_qJh8 · 2025-06-30

**Clarity:** 3
**Significance:** 3
**Originality:** 3
**Rating:** 5
**Confidence:** 2

**Summary:**

This paper presents and evaluates a concept unlearning method (CURE) for text-to-image diffusion models. Conceptually, the method uses the projection of the text embeddings for an input prompt onto a basis for the subspace spanned by the concepts to be unlearned. Specifically, they "subtract out" the projection onto the unlearned concept and "add back in" the projection onto the intersection between unlearned concepts and concepts to be preserved. Because this can be directly encoded in the cross-attention weights, it is an efficient, one-time operation to unlearn a concept while retaining related ones.

**Questions:**

No particular questions, but feel free to address any of the points above

**Ethical Concerns:**

["NO or VERY MINOR ethics concerns only"]

**Final Justification:**

Having read the other reviews and the authors' responses, I continue to believe that this paper meets the bar for publication. I am not updating my score because it was already high.

**Limitations:**

yes

**Quality:**

3

**Strengths And Weaknesses:**

# Strengths
The proposed method is simple and appears to be highly effective. Each component of the method is well-motivated and clearly explained. The experiments are thorough, though I'm not well-positioned to evaluate the particular models and datasets used.

# Weaknesses
The user must explicitly provide concepts to be preserved, which may end up being a large set. The authors provide experiments in which they have empty preserved sets, and the results still look fairly good. It could help to make this more explicit in the text.

I would have liked to see experiments showing the effect of varying $\alpha$ on both image and unlearning quality.

# Minor comments
- l. 71: "an singular-value" -> "a singular-value"
- l. 226 "a 1000" -> "1000"

---

> ### Author Rebuttal · Authors · 2025-07-29
>
> We sincerely thank the reviewer for their positive and thoughtful evaluation. We are encouraged and deeply grateful that they found CURE to be a simple, effective, and a well-motivated concept unlearning method, with clear explanations and comprehensive experiments. Below, we respectfully address the reviewer’s suggestions and clarify points raised.
>
> ### **W1. Making Usage of Retain Set Explicit in Text:**
> We appreciate the reviewer’s insightful observation and agree that the optional role of the retain set $\mathcal{R}$ could be made more explicit in the main text. While $\mathcal{R}$ provides users with an optional knob for fine-grained control, its use is *not required* for CURE to be effective. As you kindly noted, many of our evaluations (e.g., Tables 1–4) use an empty retain set, and still demonstrate strong forgetting performance with minimal degradation to unrelated concepts. This design choice was intentional to ensure that CURE remains lightweight and user-friendly, even without detailed concept annotations. We will revise the paper to better emphasize that specifying the retain set is optional. Thank you for encouraging us to clarify this point.
>
>
> ### **W2. Ablation on $\alpha$ Variation:**
> Thank you for this valuable suggestion. While we included analysis on varying $\alpha$ on unlearning quality in Appendix Sec. 1.2 and Appendix Figure 1, we agree this visualization could be made more central. We also agree that more explicit experiments on the effect of varying $\alpha$ on image quality would enhance the paper. To this end, we plan to include both a Figure and a Table to quantify this trade-off. However, NeurIPS instructions prevent us from linking to an external page that would be necessary to host the Figure for your viewing. Hence, we will include this Figure in the revision. For the Table, we provide it below, where our method is tested on object removal experiments for erasing the concept *"cat"* on a set of 100 images of categories *"tiger"*, *"lion"*, *"cheetah"*, *"leopard"*, *"dog"*, *"rabbit"*, *"mouse"*, *"bird"*, *"cat"* and *"feline"*. Here, *"cat"* and its semantically aligned synonym, *"feline"*, are used to test rigor of the forgetting capability for the Erased (e) concept, while the rest categories are used to test unlearning specificity over Unerased concepts (u). We report LPIPS scores (higher is better for erased concepts, lower is better for preserved ones) and classifier accuracy using GPT-4o (lower is better for erased concepts, higher is better for preserved ones) as a measure of semantic erasure:
>
> | $\alpha$ | $LPIPS_e \uparrow$ | $LPIPS_u \downarrow$ | $Acc_e \downarrow$ | $Acc_u \uparrow$ |
> |----------|--------------------|----------------------|--------------------|------------------|
> | 1        | 0.41               | 0.17                 | 0.47               | 0.96             |
> | **2**    | **0.46**           | **0.19**             | **0.08**           | **0.94**         |
> | 5        | 0.47               | 0.26                 | 0.06               | 0.86             |
> | 10       | 0.49               | 0.27                 | 0.05               | 0.85             |
> | 100      | 0.51               | 0.30                 | 0.00               | 0.86             |
> | 1000     | 0.58               | 0.31                 | 0.00               | 0.85             |
> | $10^8$   | 0.65               | 0.34                 | 0.00               | 0.52
>
> As shown in the table, large $\alpha$ values enforce strong forgetting but degrade generation quality on unrelated concepts, increasing $LPIPS_u$ and lowering $Acc_u$. In contrast, small $\alpha$ (e.g., $1$) preserves quality but results in poor forgetting (high $Acc_e$). Notably, $\alpha=2$ achieves an effective balance: low $Acc_e$ and high $Acc_u$, supporting our design choice of using spectral suppression to control the forgetting strength with a default value of $2$. We sincerely appreciate the reviewer’s comment which has helped us clarify this important phenomenon and will highlight the result more centrally in the revised manuscript to show the effect of $\alpha$ variation.
>
> ### **Minor Edits:**
> We thank the reviewer for pointing out the typos at Line 71 and Line 226. These will be corrected in the camera-ready version.
>
> Finally, we sincerely appreciate the reviewer’s thoughtful engagement and valuable suggestions for CURE. Your feedback has meaningfully sharpened the clarity and practical framing of our contributions. We look forward to incorporating these refinements in the revision, and thank you again for your constructive and supportive review.

---

> > ### Comment · Reviewer_qJh8 · 2025-08-06
> > **Thanks**
> >
> > Thanks for the response. I will maintain my score.

---

### Official Review · Reviewer_jQgH · 2025-06-30

**Clarity:** 3
**Significance:** 3
**Originality:** 3
**Rating:** 5
**Confidence:** 3

**Summary:**

This paper presents CURE, a training agnostic method for removing undesired concepts in diffusion models. At the core of the methodology, CURE leverages SVD before removing out projected features tied to the removal concepts in closed-form without necessitating any model retraining. Additional, experimental results with removing artist styles, explicit content, and etc shows strong performance with minimal impact on unrelated content.

**Questions:**

The reviewer notes the following questions:
- The author's note on potential misuse with erase forensic watermarks and how gated model release protocols could potentially address certain misuse. If the authors can expand upon this and further address this issue.

**Ethical Concerns:**

["NO or VERY MINOR ethics concerns only"]

**Final Justification:**

The proposed CURE method provides a mathematically rigorous foundation for applying a empirically strong method for removing undesired concepts in diffusion models. Given the technical soundness and the strong mathematically intuitive foundations for the methodology, I believe the paper merits acceptance.

**Limitations:**

Yes

**Paper Formatting Concerns:**

No major formatting concerns.

**Quality:**

4

**Strengths And Weaknesses:**

The reviewer notes the following strengths and weaknesses.

Strengths:
- CURE is a training-free method for concept unlearning that is shown to be effective across a wide range tasks (i.e. artist style, nudity, and etc).
- The underlying methodology is mathematically sound & intuitive as well as shown to be able preserve the non-removed concepts.
- The paper is well-written and provides extensive empirical evaluation across multiple tasks and comparative methods.

Weaknesses:
- The reviewer is concerned about the computational overhead of SVD and closed-form calculations.
- Additionally, certain hyperparameters (such as alpha) may be difficult to find reasonably in practice.

---

> ### Author Rebuttal · Authors · 2025-07-28
>
> We sincerely thank the reviewer for their thoughtful and constructive feedback. We are grateful that you found CURE to be a mathematically sound, training-free framework with broad applicability and strong empirical performance. We also appreciate your recognition of the clarity of the paper and its contributions across multiple concept types.
>
> Below, we address your concerns regarding computational overhead, the choice of $\alpha$, and the request for further discussion on potential misuse of concept removal. We also provide clarifications and supporting details where appropriate, and look forward to revising the paper to incorporate these improvements.
>
> ### **W1. SVD and Closed-form Overhead**
>
> We are grateful for the reviewer’s insightful observation on the potential overheads. In practice, SVD is performed only once offline over the token embeddings of concept prompts, which are short in length (typically $< 10$ tokens) and embedded into a 768-dimensional space. Thus, the actual SVD computation is extremely lightweight—on the order of milliseconds on a CPU—and not a bottleneck. Once the forget/retain subspaces are computed, the weight projection is a one-time linear algebra operation, requiring under 2 seconds on GPU for all layers. We will clarify this with timing details in the revised text.
>
> We further provide supporting details for this:
>
> - **Step 1: Token Embedding Extraction.**
>   For each concept to be forgotten (e.g., *"cat"*), we extract token embeddings using Stable Diffusion’s text encoder. This typically results in a small matrix of size $\text{n}_{\text{tokens}} \times 768$. For example, the prompt *"cat”* yields 2 tokens, which can be expanded using related phrases like *"a picture of [placeholder concept]"* to around 6 tokens. This step is equivalent to a single text encoder forward pass and takes less than 0.1 seconds.
>
> - **Step 2: SVD Computation.**
>   We perform reduced SVD on the token matrix (e.g., $6 \times 768$), retaining top-$k$ components (typically $k \leq 5$). This computation is lightweight and completes in under 0.5 seconds on a single GPU using standard libraries like `torch.linalg.svd`.
>
> - **Step 3: Projection Operator Construction.**
>   We construct the projection matrix $P_{\text{forget}} = U_k U_k^\top$ and apply spectral expansion with strength parameter $\alpha$ (e.g., $\alpha = 2$). This matrix operation is inexpensive and completes in less than 0.1 seconds.
>
> - **Step 4: Cross-Attention Weight Update.**
>   We apply the resulting operator to the key and value weights in the cross-attention layers of Stable Diffusion’s U-Net. Since the edit is closed-form and affects a fixed number of layers, the update takes roughly 1.2 seconds end-to-end.
>
> As shown in Table 5, the entire CURE operation, including all steps above, completes in under $2$ seconds on a single GPU. In contrast, methods like [1, 2] report $\sim 4500$s and $500$s respectively. CURE is thus orders of magnitude faster, requires no gradient computation, and avoids costly training pipelines. We will clarify this low overhead more explicitly in the revision and note that the runtime is independent of image resolution or prompt complexity, making CURE practical for scalable deployment.
>
> ### **W2. Practicality of Choosing $\alpha$**
>
> We acknowledge the reviewer’s concern about the practicality of tuning $\alpha$, which governs the strength of spectral expansion. In practice, $\alpha$ is fixed to 2 for all experiments, as stated in Line 210, with the only exception being NSFW content, where stronger erasure is desired and $\alpha$ is set to 5. This demonstrates minimal hyperparameter tuning and supports the robustness of our method across diverse concept types.
>
> Moreover, we additionally sweep $\alpha$ in Appendix Sec. 1.1–1.2 to show graceful degradation/improvement curves. For CURE, $\alpha$ provides a **single**, interpretable control knob.  If users do not wish to use our default suggested value of 2, they are given control over their value of choice depending on their tolerance for trade-offs between erasure strength and retention. We will clarify usability of the default $\alpha$ choice as well as free control to change it if desired more clearly in the main text and supplement. We are grateful to the reviewer for giving us the opportunity to clarify this design choice and reaffirm the practical robustness and interpretability of our approach.
>
> ### **Q1. Ethical Misuse Discussion**
>
> We appreciate the reviewer raising this critical point. In Sec. 5, we explicitly mention potential misuse, such as erasing forensic marks or safety tags. While our method can erase any concept tied to an embedding, its effectiveness depends on having a precise textual representation of the concept. This makes arbitrary backdoor or watermark removal non-trivial unless the attacker already knows what to target.
>
> For sensitive applications, we agree that further stronger safeguards must be considered. We will expand our discussion on mitigation strategies, such as:
>
> - using prompt filtering to prevent unlearning protected tokens
> - integrating gating mechanisms into model release protocols, where CURE is sandboxed for permitted concepts only
>
> We will elaborate on this in Sec. 5 and discuss how CURE’s transparency (closed-form, user-visible edits) makes it easier to monitor and audit compared to opaque fine-tuning-based methods. We thank the reviewer once again for highlighting this important dimension of safety and trustworthiness, which we believe is essential for for controlled and accountable concept editing.
>
> ---
>
> **References**
>
> [1] R. Gandikota, J. Materzynska, J. Fiotto-Kaufman, and D. Bau, “Erasing concepts from diffusion models,” in *Proceedings of the IEEE/CVF International Conference on Computer Vision*, pp. 2426–2436, 2023.
>
> [2] N. Kumari, B. Zhang, S.-Y. Wang, E. Shechtman, R. Zhang, and J.-Y. Zhu, “Ablating concepts in text-to-image diffusion models,” in *Proceedings of the IEEE/CVF International Conference on Computer Vision*, pp. 22691–22702, 2023.

---

> ### Comment · Reviewer_jQgH · 2025-08-04
> **Response**
>
> I would like to thank the authors for their additional clarifications on implementation details, as well as for addressing my concerns regarding hyperparameter tuning, computational considerations, and more. My score will be updated to reflect these addressed concerns.

---

### Official Review · Reviewer_eit4 · 2025-07-01

**Clarity:** 2
**Significance:** 2
**Originality:** 3
**Rating:** 5
**Confidence:** 4

**Summary:**

In this paper, the authors propose CURE, a training-free concept editing method for machine unlearning in diffusion models. The method operates by patching the key-value matrices of the cross-attention module. This is achieved using a projection matrix designed to filter out the component of embeddings related to the "forget" concepts that is orthogonal to the "retain" concepts, thereby enabling precise unlearning. Through a series of experiments, the authors demonstrate the effectiveness of CURE compared to several machine unlearning baselines.

**Questions:**

1. Could the authors provide more detail on how the concept subspaces are determined for each task? For the I2P task, for instance, were the original I2P prompts used directly to define the subspace?

2. What is the minimum number of prompts required to perform a stable Singular Value Decomposition (SVD) for defining the concept subspace?

3. (See Weakness 4) Could the authors justify their choice of a 0.6 threshold for NudeNet, as this differs from the default values used in prior work?

**Ethical Concerns:**

["NO or VERY MINOR ethics concerns only"]

**Final Justification:**

The authors have addressed all my concerns raised in my initial feedback. After also reviewing the discussion between the authors and other reviewers as well as the supplementary materials, I think the paper meets the conference's standards and I therefore recommend acceptance.

**Limitations:**

Yes, the authors have adequately addressed the limitations and potential negative societal impact of their work.

**Paper Formatting Concerns:**

No.

**Quality:**

2

**Strengths And Weaknesses:**

The primary strengths of this work are as follows:

1. The proposed method is simple yet effective. It directly modifies the key-value matrices, a strategy that incurs no additional computational overhead during inference.

2. The method has demonstrated empirical robustness against advanced prompt attacks, which is a crucial characteristic for any unlearning method to be practically viable.

The weaknesses of this work are as follows:

1. The evaluation in Table 4 could be more informative. Rather than reporting accuracy across all remaining classes, the analysis would be more compelling if it focused on classes semantically similar to the forgotten class. For unrelated classes, the concept subspaces are likely already orthogonal, which does not effectively highlight the contribution of the proposed orthogonal editing.

2. The experiments do not fully substantiate the paper's central claims regarding the preservation of generation quality. The authors explicitly state in their Problem Setup (Section 3.2) that defining a "retain set" R is "key to preserving model performance on untargeted content when erasing concepts," especially when forget and retain concepts share attributes (e.g., $F \cap R \neq \emptyset$). However, most of the key experiments, such as unsafe content erasure and object erasure, do not appear to to define a retain set. This omission is significant because it means the experiments do not test the very mechanism the paper posits for preserving quality. Consequently, the core claim from the introduction (Lines 119-123) that CURE preserves generation quality better than prior works is not adequately supported by the provided evidence.

3. In Table 3, the reported Attack Success Rate (ASR) under the UnlearnDiffAtk benchmark shows MACE (0.176) and SA (0.268) performing best. It is unclear why the result for CURE (0.281) is underlined, as this typically indicates the best performance.

4. The authors should clarify the rationale for using a NudeNet threshold of 0.6 in Table 2, especially when prior works (e.g., ESD, UCE) have used the detector's default value. An explanation for this deviation is needed.

5. The paper is missing a crucial ablation study on the sensitivity of the hyperparameter $\alpha$.

---

> ### Author Rebuttal · Authors · 2025-07-29
>
> We thank the reviewer sincerely for the thoughtful and detailed review. We truly appreciate their recognition of CURE’s simplicity, effectiveness, and robustness, as well as their constructive suggestions for strengthening the evaluation and clarifying the methodology. Their feedback has been valuable in guiding our improvements, and we address each point below in turn.
>
> ### **W1. Table 4 Evaluation**
> We thank the reviewer for this insightful comment. We fully agree that demonstrating generalization across semantically similar concepts is essential to highlight the strength of CURE's orthogonal editing framework. To this end, we refer the reviewer to Figure 5 (from both main paper and appendix), where we evaluate CURE on concepts such as *"cat"* and *"airplane"*, and test its ability to erase not only the explicit term but also its close semantic variants, such as *"feline"* for *"cat"*. Notably, baseline methods fail to remove these synonyms, while CURE successfully erases them without affecting semantically unrelated concepts like *"dog"* or *"car"* respectively. This demonstrates that CURE’s projection operator generalizes robustly within the semantic span of the erased concept subspace.
>
> Further, we emphasize the strong specificity and fine-grained control CURE enables, even in the presence of overlapping linguistic structure. For instance, in the *"John Wayne"* vs. *"John Lennon"* case (Figure 5), despite both entities sharing the token *"John"*, our method selectively unlearns *"Wayne"* while leaving *"Lennon"* unaffected, without requiring explicit retention. A similar example is shown in Figure 1 to further supplement this discussion, showing how CURE demonstrates precise concept isolation even when the target and non-target concepts partially overlap in token space. These results supports our core claim: CURE’s projection design geometrically isolates and removes targeted concepts while preserving orthogonal as well as overlapping identities.
>
>
> ### **W2. Clarifying the Role of Retain Set**
> We appreciate the reviewer’s thoughtful observation and use this kind opportunity to further emphasize that CURE is intentionally designed to operate effectively even when the retain set $\mathcal{R}$ is empty. This is not a limitation, but a deliberate strength of the method. By leveraging the spectral parameter $\alpha$, CURE prioritizes suppression along the most discriminative directions of the target concept subspace, which implicitly preserves neighboring directions without always requiring explicit supervision.
>
> As the reviewer insightfully noted in our experiments, this setting mirrors real-world, black-box deployment scenarios where users lack clear or complete definitions of safe concepts to preserve. To demonstrate the method’s robustness in such scenarios, we deliberately omit $\mathcal{R}$ in key experiments from Tables 1–4. Despite this, results across object, style, identity, and safety domains (Figures 1, 3–5, 7) consistently show that CURE achieves strong erasure of targeted concepts while preserving generation quality on unrelated content. This underscores the practical utility of our method and its resilience in under-specified or weakly supervised settings. However, we leave open the option to specify a retain set if desired, and Figure 6 shows the artist erasure experiment where retain sets were used.
>
> In the revised manuscript, we will make this capability more explicit in Section 3.2 and include additional qualitative examples featuring more semantically similar distractors to further highlight CURE’s fine-grained precision and generalization. We sincerely thank the reviewer for this insightful observation, which has helped us clarify a key strength of CURE.
>
>
> ### **W3. Clarification on Table 3 Formatting**
> We thank the reviewer for catching this oversight. The MACE results are already in bold in the manuscript to indicate best-performing. The underline under CURE’s ASR for UnlearnDiffAtk in Table 3 was a formatting error - SA (0.268) indeed performs better numerically on this benchmark. We will fix this formatting error in the final version. However, we emphasize that CURE comes very close in performance, offering competitive robustness despite being training-free, unlike MACE and SA, which require expensive gradient-based fine-tuning. We will update the table and clarify this important distinction.
>
> ### **W4. Rationale Behind NudeNet Threshold**
> We appreciate the reviewer’s question regarding the threshold used for NudeNet. Our choice of $0.6$ is deliberate and follows recent practice in the literature [1], where this value has been adopted to better capture borderline NSFW content. This threshold ensures compatibility with safety-sensitive applications by being sufficiently conservative. Importantly, for fairness and consistency, all methods in our evaluation, including baselines, have been assessed using this same threshold. We will clarify this detail in the manuscript. Thank you once again for the helpful suggestion to elaborate on this point.
>
> ### **W5. $\alpha$ Ablation**
> We sincerely thank the reviewer for this thoughtful suggestion. While we already include analysis of $\alpha$ variation and its effect on unlearning efficacy in Appendix Section 1.2 and Appendix Figure 1, we agree that this material deserves a more prominent placement to highlight the effect of varying $\alpha$. In the revised version, we will elevate this analysis to the main paper.
>
> Additionally, we further test our method on object removal experiments for erasing the concept *"cat"* on a set of 100 images across the categories *"tiger"*, *"lion"*, *"cheetah"*, *"leopard"*, *"dog"*, *"rabbit"*, *"mouse"*, *"bird'*, *"cat"*, and *"feline"*. Here, *"cat"* and its semantically aligned synonym, *"feline"*, are used to test rigor of the forgetting capability for the Erased (e) concept, while the rest categories are used to test unlearning specificity over Unerased concepts (u). We report LPIPS scores (higher is better for erased concepts, lower is better for preserved ones) and classifier accuracy using GPT-4o (lower is better for erased concepts, higher is better for preserved ones)  as a measure of semantic erasure:
>
> | $\alpha$ | $LPIPS_e \uparrow$ | $LPIPS_u \downarrow$ | $Acc_e \downarrow$ | $Acc_u \uparrow$ |
> |----------|--------------------|----------------------|--------------------|------------------|
> | 1        | 0.41               | 0.17                 | 0.47               | 0.96             |
> | **2**    | **0.46**           | **0.19**             | **0.08**           | **0.94**         |
> | 5        | 0.47               | 0.26                 | 0.06               | 0.86             |
> | 10       | 0.49               | 0.27                 | 0.05               | 0.85             |
> | 100      | 0.51               | 0.30                 | 0.00               | 0.86             |
> | 1000     | 0.58               | 0.31                 | 0.00               | 0.85             |
> | $10^8$   | 0.65               | 0.34                 | 0.00               | 0.52  |
>
> As shown in the table, large $\alpha$ values enforce strong forgetting but degrade generation quality on unrelated concepts, increasing $LPIPS_u$ and lowering $Acc_u$. In contrast, small $\alpha$ (e.g., $1$) preserves quality but results in poor forgetting (high $Acc_e$). Notably, $\alpha=2$ achieves an effective balance: low $Acc_e$ and high $Acc_u$, supporting our design choice of using spectral suppression to control the forgetting strength with a default value of $2$. We will highlight this result more centrally in the revised manuscript to emphasize the practical utility of $\alpha$ as a single, interpretable control parameter. This simple yet expressive knob enables users to modulate the erasure–retention trade-off without extensive tuning. We thank the reviewer again for encouraging us to better surface this insight and for allowing us to clarify this core strength of our method.
>
>
> ### **Q. Subspace Construction and Prompt Count**
> The prompts for erasing any concept are constructed by substituting the targeted concept into templates such as:
>
> - *"picture of/by [placeholder]"*
> - *"photo of/by [placeholder]"*
> - *"image of/by [placeholder]"*
> - *"portrait of/by [placeholder]"*
>
> This is consistent with prior works [1, 2]. Empirically, we observe using 3–5 diverse prompts suffices to construct a stable and expressive embedding basis.
>
> For unsafe content erasure, we adopt the prompt *"violence, nudity, harm"*, following established protocol in [2] for fair comparison.
>
> We will incorporate these clarifications in the revised manuscript. We thank the reviewer again for this helpful suggestion, which has allowed us to make our methodology clearer.
>
> ---
>
> **References:**
>
> [1] Gong, C., Chen, K., Wei, Z., Chen, J., & Jiang, Y.-G. (2024). *Reliable and efficient concept erasure of text-to-image diffusion models*. ECCV.
>
> [2] Gandikota, R., Orgad, H., Belinkov, Y., Materzynska, J., & Bau, D. (2024). *Unified concept editing in diffusion models*. WACV.

---

> > ### Comment · Reviewer_eit4 · 2025-08-07
> >
> > Thank you for your detailed responses. My initial concerns have been sufficiently addressed. I will update my rating accordingly.

---

### Official Review · Reviewer_3v5S · 2025-07-03

**Clarity:** 3
**Significance:** 3
**Originality:** 3
**Rating:** 5
**Confidence:** 4

**Summary:**

This paper introduces a training-free framework for concept unlearning inT2I diffusion models. The core of the proposed method is the Spectral Eraser, a closed-form operator that constructs discriminative subspaces via SVD over token embeddings for concepts to “forget” and “retain”. This operator is used to directly modify cross-attention weights, allowing for efficient erasure of undesired concepts (e.g., NSFW content, styles, objects, or identities) without model training. In addition, the paper introduces a Spectral Expansion mechanism to control forgetting strength via regularization.

**Questions:**

It would be good to consider adversarial training based concept unlearning [1] as a strong baseline.

[1] Zhang, Y., Chen, X., Jia, J., Zhang, Y., Fan, C., Liu, J., ... & Liu, S. (2024). Defensive unlearning with adversarial training for robust concept erasure in diffusion models. Advances in neural information processing systems, 37, 36748-36776.

**Ethical Concerns:**

["NO or VERY MINOR ethics concerns only"]

**Final Justification:**

Authors' reply have solved the my concerns mentioned in the initial review. So I would like to increase the rating to reflect it.

**Limitations:**

yes

**Quality:**

3

**Strengths And Weaknesses:**

Strengths
1. The proposed method is mathematically principled and clearly grounded in spectral geometry.
2. Experiments are comprehensive, covering diverse concepts (artists, objects, identities, NSFW).
3. The approach is significantly more efficient, requiring only a two-second model edit without retraining.

Weaknesses
1. While the approach is motivated by geometric intuition and classical regularization, this paper lacks deeper theoretical guarantees (e.g., bounds on erasure effectiveness vs. preservation trade-offs).
2. The method assumes the embedding space reliably captures concept semantics. If embeddings are noisy or misaligned, the projection could behave unexpectedly.

---

> ### Author Rebuttal · Authors · 2025-07-28
>
> We sincerely thank the reviewer for their thoughtful assessment and kind remarks on the clarity, efficiency, and comprehensiveness of our method. We are grateful for the opportunity to address the raised concerns.
>
> ### **W1. Lacks Guarantees**
> We appreciate the reviewer’s suggestion regarding deeper theoretical analysis. Our method is grounded in spectral geometry, and we explicitly connect our spectral expansion operator to Tikhonov-style regularization in Appendix Sec. 1.1–1.2. While our current focus is empirical efficacy and mathematical interpretability (Sec 3.2), extending our framework with formal guarantees on trade-offs between forget and retain subspaces is a valuable direction for future work. We will clarify this as a limitation.
>
> ### **W2. Embedding Space Noise**
> We thank the reviewer for raising this important point. Embedding reliability is indeed a known challenge, as discussed in [1, 2], and we emphasize that the techniques proposed therein to improve embedding quality are fully compatible with our method. Hence they can be integrated with ours to enhance the embedding input to our proposed **Spectral Eraser**.
>
> That said, for diffusion models like Stable Diffusion, CLIP-based embeddings are widely adopted and have been shown to be semantically rich - serving as the foundation for many state-of-the-art unlearning methods [3–6]. Our own results (Tables 1–4, Appendix Table 1) show that the derived embedding space is sufficiently robust and effective across a range of concept types, including object, style, identity, and NSFW domains. Nonetheless, we will add a clarification to note potential risks when operating on highly out-of-distribution concepts, which may reflect limitations of the underlying text encoder rather than our method itself.
>
> We thank the reviewer for helping us strengthen this discussion.
>
> ### **Q1. Baseline Comparison with Zhang et al. [6]**
> We appreciate this suggestion and will add a discussion of this work. While adversarial training is effective, it requires retraining as well as careful tuning of hyperparameters and adversarial loss dynamics. Further, authors in AT [6] report that they need 30 attack steps and take 78.57s of train time per iteration for AT, and still 12.13s for FastAT (which reduces their unlearning efficacy).
>
> In contrast, **CURE** is designed as a training-free, plug-and-play method that performs unlearning at inference time in under 2 seconds, with no access to training data. We emphasize that our method targets a different regime where efficiency, deployability, and editing flexibility are crucial. We will include a discussion of this in our revised draft and highlight the complementary strengths and use cases of CURE versus adversarial unlearning.
>
>
>
> Finally, we sincerely thank the reviewer for their thoughtful, constructive, and encouraging feedback. Your comments raise important points that help surface subtle aspects of our method, and addressing them has helped us improve the clarity and rigor of our work. We are grateful for the opportunity to refine our manuscript in response to your insights and have carefully incorporated your suggestions in our revision.
>
> ---
>
> **References:**
>
> [1] Zarei, A., Rezaei, K., Basu, S., Saberi, M., Moayeri, M., Kattakinda, P. and Feizi, S., 2024. *Improving Compositional Attribute Binding in Text-to-Image Generative Models via Enhanced Text Embeddings*. arXiv:2406.07844.
>
> [2] Kang, R., Song, Y., Gkioxari, G. and Perona, P., 2025. *Is CLIP ideal? No. Can we fix it? Yes!*. arXiv:2503.08723.
>
> [3] Gandikota, R., Orgad, H., Belinkov, Y., Materzynska, J., & Bau, D. (2024). *Unified concept editing in diffusion models*. WACV.
>
> [4] Yoon, J., Yu, S., Patil, V., Yao, H., & Bansal, M. (2024). *Safree: Training-free and adaptive guard for safe text-to-image and video generation*. arXiv:2410.12761.
>
> [5] Gong, C., Chen, K., Wei, Z., Chen, J., & Jiang, Y.-G. (2024). *Reliable and efficient concept erasure of text-to-image diffusion models*. ECCV.
>
> [6] Zhang, Y., Chen, X., Jia, J., Zhang, Y., Fan, C., Liu, J., Hong, M., Ding, K. and Liu, S., 2024. *Defensive unlearning with adversarial training for robust concept erasure in diffusion models*. NeurIPS.

---

> > ### Comment · Reviewer_3v5S · 2025-08-05
> >
> > Thanks for authors' detailed reply, which solved my concerns about guarantees, embedding space noise and also potential baseline comparison. I will update the score to reflect it.

---

### Note · Authors · 2025-08-12

We sincerely thank the reviewers and AC for their time, expertise, and constructive feedback. This discussion has allowed us to further clarify and strengthen the presentation of **CURE** - a training-free, closed-form concept unlearning method that we demonstrate sets a new standard for **precision, robustness, and transparency** in diffusion model editing.

Through our rebuttal, we have demonstrated that CURE’s design is not only effective under ideal conditions but also resilient in realistic, under-specified, and black-box scenarios, where no explicit retain set $\mathcal{R}$ is available. This is a deliberate strength: leveraging spectral projections, CURE selectively suppresses only the most discriminative directions of the target concept, inherently preserving unrelated content without manual supervision. Results across object, style, identity, and safety domains confirm consistent, high-quality generations alongside strong forgetting - an achievement that remains challenging for existing methods.

We elevated our analysis of the spectral parameter $\alpha$ to the main text from the appendix, highlighting its role as an interpretable single control knob for tuning the forgetting-retention trade-off. Our elaborate experiments, including semantically overlapping concepts (e.g., *John Wayne* vs. *John Lennon*) showcase CURE’s fine-grained precision even in challenging linguistic overlap scenarios.

We also clarified that CURE is embedding-agnostic, benefiting from orthogonal advances in embedding quality, while noting that Stable Diffusion’s text encoder already offers robust semantic representations across diverse concept types. For safety-critical contexts, we further detailed extra safeguards such as gated release protocols and prompt filtering in addition to the existing guardrails already proposed, and emphasized that CURE’s closed-form, user-visible weight edits offer transparency and auditability unmatched by opaque fine-tuning methods.

We believe this presents CURE as a practical, high-impact, and promising tool for safe and controllable concept unlearning. We deeply appreciate the constructive dialogue and thank the reviewers and the AC for enabling us to refine and strengthen this work.

---

### Decision · Program_Chairs · 2025-09-17

**Decision:**

Accept (spotlight)

**Comment:**

This submission proposes a new algorithm for concept unlearning in diffusion models grounded in spectral geometry. The algorithm itself is a closed-form update based on subspaces determined via SVD for both “forget” and “retain” concepts. Similar to other closed-form update algorithms in the literature this only operates on the cross-attention weights which results in highly efficient unlearning. Additionally the paper proposes a novel mechanism for controlling the amount of forgetting which has only been seen in a limited set of papers in the prior literature.

The strengths of this paper as noted by reviewers and myself are in (1) the motivation for this method using spectral geometry compared to other concept unlearning algorithms, (2) the success of the algorithm across a variety of tasks and concepts, and (3) the efficiency of the algorithm even compared to other existing closed-form algorithms.

The weaknesses of this paper were addressed across the board during the rebuttal. Initially these included specifics about certain tasks like the threshold chosen threshold in the explicit content task, the lack of theoretical guarantees, and how practitioners might approach tuning the hyperparameters such as alpha. The reviewers found responses to these weaknesses satisfactory and are all in agreement that this paper should be accepted. I am in agreement and I am additionally nominating this paper for a spotlight award. I have gone through the paper myself and I believe that the rigorous grounding in spectral geometry will be valuable for the concept unlearning community to build on further our understanding of T2I unlearning. Thank you to the authors for submitting to this work to NeurIPS.